# Necroptosis in Organ Transplantation: Mechanisms and Potential Therapeutic Targets

**DOI:** 10.3390/cells12182296

**Published:** 2023-09-17

**Authors:** Yajin Zhao, Kimberly Main, Tanroop Aujla, Shaf Keshavjee, Mingyao Liu

**Affiliations:** 1Latner Thoracic Surgery Research Laboratories, Toronto General Hospital Research Institute, University Health Network, Toronto, ON M5G 1L7, Canada; yajin.zhao@utoronto.ca (Y.Z.); kimberly.main@uhn.ca (K.M.); tanroop.aujla@uhn.ca (T.A.); shaf.keshavjee@uhn.ca (S.K.); 2 Institute of Medical Science, Temerty Faculty of Medicine, University of Toronto, Toronto, ON M5S 1A8, Canada; 3 Department of Physiology, Temerty Faculty of Medicine, University of Toronto, Toronto, ON M5S 1A8, Canada; 4 Department of Surgery, Temerty Faculty of Medicine, University of Toronto, Toronto, ON M5T 1P5, Canada

**Keywords:** programmed cell death, donor organ condition, ischemia–reperfusion injury, allograft rejection, inflammation

## Abstract

Organ transplantation remains the only treatment option for patients with end-stage organ dysfunction. However, there are numerous limitations that challenge its clinical application, including the shortage of organ donations, the quality of donated organs, injury during organ preservation and reperfusion, primary and chronic graft dysfunction, acute and chronic rejection, infection, and carcinogenesis in post-transplantation patients. Acute and chronic inflammation and cell death are two major underlying mechanisms for graft injury. Necroptosis is a type of programmed cell death involved in many diseases and has been studied in the setting of all major solid organ transplants, including the kidney, heart, liver, and lung. It is determined by the underlying donor organ conditions (e.g., age, alcohol consumption, fatty liver, hemorrhage shock, donation after circulatory death, etc.), preservation conditions and reperfusion, and allograft rejection. The specific molecular mechanisms of necroptosis have been uncovered in the organ transplantation setting, and potential targeting drugs have been identified. We hope this review article will promote more clinical research to determine the role of necroptosis and other types of programmed cell death in solid organ transplantation to alleviate the clinical burden of ischemia–reperfusion injury and graft rejection.

## 1. Introduction

Over the past several decades, transplantation has been performed for all the major solid organs and has become the most effective therapy for patients with end-stage organ failure [1]. However, the current practice continues to face major challenges, including shortages in organ donation and the quality of donated organs. The latter is determined by the donor conditions (e.g., age, sex, smoking history, obesity, and organ damage before donation), the types of donations (e.g., donation after brain death (DBD) vs. donation after circulatory death (DCD)), organ preservation conditions (e.g., temperature, time, static storage vs. machine perfusion), and reperfusion conditions (e.g., anesthesia, surgery, post-operative care). Graft quality is linked to post-operative primary graft dysfunction (PGD), which is a major cause of early morbidity and mortality after organ transplantation [2]. It also contributes to acute and chronic rejection post-transplantation. Moreover, the increased incidence and severity of infection (viral, bacterial, or fungal), the transfer of cancer from donor to recipient, and carcinogenesis are also challenges for transplant recipients. Among these pathological processes, acute and chronic inflammation and diverse types of cell death are two major underlying mechanisms for graft injury. Thus, therapeutics targeting these processes may prevent or reduce organ injury, leading to improved post-transplant outcomes.

Ischemia–reperfusion (IR) injury is an inevitable consequence of donor organ preservation and transplantation. IR injury is also recognized as one of the primary causes of PGD [3,4,5]. Cellular damage resulting from IR is an important risk factor not only for PGD but also for acute and chronic rejection [6]. It consists of complex pathophysiological processes, including endothelial and epithelial cell dysfunction, acute inflammation, and the activation of innate and adaptive immune responses. IR-induced cell death can be induced by a loss of energy supply, the elaboration of inflammatory mediators and toxic molecules, and the activation of programmed cell death (PCD). Recently, multiple different types of inflammation-related PCD (e.g., necroptosis, pyroptosis, ferroptosis, autophagy-associated cell death) have been reported in IR injury in lung transplantation [7]. Among these types of PCD, necroptosis is particularly interesting, as it has been investigated in all major solid organ transplantations with clinical relevance.

Necroptosis is defined as cell death mediated through a pathway that depends on the Receptor-Interacting Serine/Threonine-Protein Kinase 1/3 (RIPK1/3) complex (also called the necrosome), which can be induced by a class of death receptors (DRs), including TNF receptor 1 (TNFR1), TNF-related apoptosis-inducing ligand-receptors (TRAIL-Rs), Fas, and toll-like receptors (TLRs) [8,9,10,11,12]. The release of inflammatory cytokines, pathogen-associated molecular patterns, and damage-associated molecular patterns (DAMPs) in the donor and recipient graft can activate these DRs. Upon activation, the necrosome then recruits mixed lineage kinase domain-like protein (MLKL), which can be phosphorylated by RIPK3 [13] and subsequently translocated to lipid rafts in the plasma membrane, ultimately resulting in cell membrane rupture and cell death [10,11,14]. Necroptotic cells further release DAMPs, such as high mobility group box 1 (HMGB1), and cytokines to propagate the inflammatory response to surrounding cells [15]. Necroptosis can also be initiated by a range of intrinsic factors, such as reactive oxygen species (ROS) and intracellular Ca^2+^ overload [16], which, through p53 and cyclophilin D (CypD), exerts control over mitochondria permeability transition pore (mPTP) opening to trigger regulated necrosis (Figure 1) [17]. Necroptosis is involved in many pathological conditions and diseases [16,18], especially in IR injury [19].

In this review article, we summarized the scientific literature on the prevalence of necroptosis in major solid organ transplantation with a focus on the regulatory mechanisms of necroptosis discovered from organ transplantation studies. Lastly, we introduced potential therapeutic targets for necroptosis and proposed future directions for PCD-related research on organ transplantation.

## 2. Materials and Methods

For this narrative review, we conducted a literature search using PubMed and Web of Science databases up to June 2023. Our search included specific combinations of keywords such as “transplantation” or “transplant” with “necroptosis”, “RIPK1/3”, and “MLKL”. We also included “ischemia-reperfusion injury”, “donor preservation” or “graft dysfunction” along with organ-specific terms like “liver”, “kidney”, “lung”, and “heart”. Studies focusing on necroptosis in other diseases were excluded from this review.

## 3. Necroptosis in Solid Organ Transplantation

Necroptosis can be induced in the donor organ, further enhanced by IR injury, and play an important role in allograft rejection.

### 3.1. Necroptosis in Kidney Transplantation

The first evidence of necroptosis in organ transplantation was derived from rodent kidney studies. In 2013, a research group led by Dr. Zhu-Xu Zhang and Dr. Anthony Jevnikar at the University of Western Ontario, in collaboration with Dr. Linkermann (an expert in necroptosis research), demonstrated that *Ripk3*^−/−^ mice are resistant to renal IR injury and exhibited better renal function, less necrosis, and lower levels of HMGB-1 (a chromatin protein that can be secreted by immune cells or released by dead cells) in kidney tissue. *Ripk3*^−/−^ kidney allografts also had reduced histological injury scores, neutrophil infiltration, fibrosis, tubulitis, and vascular injury. Moreover, animals receiving *Ripk3*^−/−^ kidney allografts achieved greater rejection-free survival [20]. As RIPK1/3 are the most important mediators of necroptosis, this study built the foundation for further investigations on the role of necroptosis in solid organ transplantation.

Cold ischemia (CI) is commonly induced to prolong graft viability following donor organ harvest until the commencement of transplantation. However, prolonged CI is a risk factor for post-transplant acute kidney injury. Jain et al. used inbred mice to study the effects of CI on kidney transplantation. After 1 h of CI, the right kidney was transplanted to a recipient, and 7 days later, the native left kidney was removed and the function of the right renal graft at postoperative day 8 was assessed [21]. After transplantation, the acute tubular necrosis score was significantly higher in CI-preserved than directly transplanted renal grafts. This was accompanied by the increased expression of DRs including TNFR1 and TLR4 in the CI-preserved kidney. Additionally, the expression of RIPK1, RIPK3, and phosphorylated MLKL (pMLKL) was significantly higher in CI-preserved kidneys after transplantation [21]. These results indicate that donor kidney preservation conditions may influence the pathogenesis of necroptosis and post-transplant renal function.

Hypothermic machine perfusion (HMP) is a method used to optimize donor kidney quality during the cold preservation period. Unlike simple CI, HMP provides continuous perfusion and oxygenation, which can better sustain organ function and reduce preservation-related injury during transplantation. In rabbit kidneys, 25 min of warm ischemic exposure to simulate clinical DCD conditions was followed by either CI static preservation or HMP at 4–8 °C for 4 h. One day later, renal function was assessed, and HMP significantly inhibited the markers of inflammation, apoptosis, and necroptosis in donor kidneys compared to those preserved statically [22].

### 3.2. Necroptosis in Heart Transplantation

Zhang and Jevnikar’s group have also extensively studied necroptosis in heart transplantation. To investigate the effects of necroptosis on post-transplant rejection, B6 *Ripk3^−/−^* donor hearts were heterotopically transplanted into fully major histocompatibility complex (MHC) mismatched recipient BALB/c mice. Twelve days after transplantation, *Ripk3^−/−^* heart grafts showed significantly lower levels of endothelial damage, lymphocyte infiltration, necrosis, and HMGB1 compared to those from wild-type mice. Moreover, a brief immunosuppressive treatment with sirolimus markedly prolonged *Ripk3^−/−^* cardiac allograft survival [23]. They further studied the role of RIPK3 in chronic cardiac allograft rejection and demonstrated that RIPK3 deficiency protected donor cardiac grafts from CD4^+^ T cell-mediated chronic rejection and improved graft survival. Using a cell culture model, they also demonstrated that CD4^+^ T cells can induce RIPK3-dependent necroptosis in microvascular endothelial cells (MVEC) by releasing TNFα and inducing direct cell–cell contact cytotoxicity. Furthermore, Fas ligand and granzyme B may contribute to activated alloreactive CD4^+^ T cell-induced necroptosis in MVECs [24].

This group further reported that the mPTP opening is an important hallmark of necroptosis in cardiac allografts [25]. The opening of mPTP is largely regulated by CypD, which has been shown to regulate both apoptotic and necrotic cell death [26]. CypD inhibition or deficiency protected MVECs from necroptosis in cell culture. Moreover, CypD-deficient cardiac grafts showed prolonged survival in mice [25]. To further elucidate the role of necroptosis in IR injury during cardiac transplant, a cold hypoxia and warm reoxygenation cell culture model was used [27]. In MVECs, after exposure to the simulated IR conditions, apoptosis-inducing factor (AIF) translocated to the nucleus, which was prevented by the RIPK1 inhibitor, necrostatin-1 stable (Nec-1s), or CypD deficiency. Overall, CypD deficiency in donor cardiac grafts significantly mitigated graft IR injury and prolonged cardiac allograft survival [27].

Another hallmark of IR injury-induced cell death in transplantation is the cellular leakage of RNA, which can interact with TLR3 on neighbouring cells and propagate inflammation to aggravate graft injury [28]. The interaction between TLR3 and RNA leads to caspase-dependent apoptosis, RIPK-dependent necroptosis, and CypD-regulated mitochondrial damage in mouse MVECs. Moreover, TLR3 deficiency protects cardiac grafts from apoptotic and necroptotic cell death and tissue damage post-transplant in a *Tlr3*^−/−^ heterotopic heart transplant model [28].

Tuuminen et al. performed an intra-abdominal heterotopic heart transplantation with fully MHC-mismatched rats, and they reported that 4 h of CI followed by 6 h of reperfusion increased the mRNA and protein expression of RIPK1/3 in rat cardiac allografts. The pre-treatment of donors and recipients with a single dose of simvastatin (an HMG-CoA reductase inhibitor) 2 h prior to allograft transplant reduced these changes and had a beneficial effect on IR injury [29].

### 3.3. Necroptosis in Liver Transplantation

IR injury is also an important contributor to graft dysfunction in liver transplantation [5]. When C57BL6 mice were subjected to warm hepatic IR (90 min ischemia/240 min reperfusion), pre-treatment with RIPK1 inhibitor necrostatin-1 (Nec-1) before ischemic onset did not attenuate IR-induced leukocyte migration, perfusion failure, and hepatocellular injury. Western blot analysis showed baseline RIPK1 expression in livers from sham-operated mice, which was reduced in IR groups. Interestingly, caspase 3 activity was significantly elevated after IR. The authors proposed that the activation of caspase 3, which can cleave RIPK1, may negatively regulate hepatic necroptosis under these specific experimental conditions [30].

Steatotic livers are prone to more severe IR injury, which has limited their usability for transplantation. In a fatty liver mouse model induced by a Western diet, steatotic livers had increased levels of RIPK1, RIPK3, and MLKL. These mice had more severe liver injury and necrosis after IR than mice on a control diet. When *Mlkl^−/−^* mice were used, although the development of hepatic steatosis was not affected, these mice had decreased hepatic neutrophil infiltration and inflammation, irrespective of diet. Moreover, *Ripk3^−/−^* or Ripk3 kinase-dead knock-in mice were protected against IR injury 24 h after reperfusion, irrespective of diet [31]. Alcohol consumption increased the levels of liver MLKL and RIPK3 in mice and corresponded with more severe liver IR injury [32].

To investigate the possible age-dependent effects of necroptosis, young (8 weeks) and aged (100 weeks) mice were subjected to liver IR (90 min ischemia/6 h reperfusion). In the aged group, there were significant increases in liver necroptosis post-IR injury [33]. Interestingly, Nec-1 was effective at reducing hepatocyte necroptosis only in aged mice, with no significant reduction in younger mice. Furthermore, IR induced endoplasmic reticulum (ER) stress in the livers of both young and aged mice, especially aged mice. The administration of an ER stress antagonist, 4-phenylbutyrate, alleviated liver IR injury in both young and aged mice. ER stress inhibition reduced hepatocyte necroptosis primarily in aged mice [33]. These studies demonstrated that pre-existing morbidities (fatty liver, aging, alcohol consumption, etc.) constitute a risk factor for necroptosis, which can be further exacerbated by IR injury during organ preservation and transplantation.

Early allograft dysfunction (EAD) following liver transplantation is a major threat to the clinical outcome of recipients [34]. In a rat isograft liver transplant model, donor livers were preserved statically for 22 h of CI followed by transplantation. One day later, significantly higher pMLKL was observed in livers with IR injury. When human samples were studied, the pMLKL score was significantly higher in the grafts of patients who developed EAD after transplantation compared to non-EAD grafts. The pMLKL score at 1 h after reperfusion and the ratio of the pMLKL score between 1 h of reperfusion and at the end of preservation were highly predictive for EAD. Human liver grafts with a high pMLKL index had significantly elevated serum levels of aminotransferases and lactate dehydrogenase 24 h after transplantation, signifying lytic cell death. Thus, the pMLKL index can serve as a reliable prognostic tool for the progression of EAD [35].

### 3.4. Necroptosis in Lung Transplantation

Interestingly, the first report on transplant-related necroptosis in the lung was from a study that investigated a phenomenon commonly referred to as kidney–lung crosstalk. In a rat allogeneic kidney transplantation model, RIPK1 and RIPK3 expression were significantly enhanced in the lung. Nec-1 given to recipients with ischemic renal grafts improved lung morphology, evidenced by decreased hemorrhage and leukocyte infiltration. The acute immune rejection of renal allograft exacerbated lung injury with enhanced RIPK1 expression. The treatment of animals with cyclosporine A, an immunosuppressive agent, significantly reduced lung injury, and when Nec-1 was combined with cyclosporine A, there was an additive benefit for lung protection [36]. Cyclosporine A is not only an immunosuppressive drug, but also an effective inhibitor of mPTP opening-related regulated necrosis [37]. It was further proposed that osteopontin, a multifunctional glycophosphoprotein that may mediate systemic inflammation, is involved in necroptosis and lung injury after the transplantation of ischemic renal allografts [38].

To determine the underlying mechanisms of IR injury in the lung transplant setting, Kim et al. used a CI and warm reperfusion cell culture model that mimicked the IR process of lung transplantation [39,40]. Human lung epithelial cells were stored at 4 °C in 50% oxygen to simulate donor lung preservation, and the cells were then returned to a serum-containing culture medium at body temperature (37 °C) to simulate warm reperfusion. Reperfusion induced mPTP opening and regulated necrosis after prolonged CI. This is mediated by the translocation of p53 into mitochondria to form a complex with CypD. Protein kinase C delta (PKCδ) plays an important role in signal transduction related to oxidative stress. The selective inhibition of PKCδ by small interference RNA (siRNA) or δV1-1, a peptide PKCδ inhibitor, reduced IR-induced inflammation, ER stress, and cell death. δV1-1 also reduced PKCδ and p53 translocation to mitochondria [39,41]. δV1-1 and its nano formula also prevented lung IR injury in a left-lung transplantation model and in a warm pulmonary IR injury model in rats [39,41].

Kim et al. further found that the RIPK1 inhibitor, Nec-1, reduced IR-induced necroptosis using the aforementioned cell culture model [40]. Necroptosis is usually triggered by the activation of DRs [18]; however, in this study, blocking DRs did not affect IR-induced necroptosis [40]. Interestingly, N-acetyl-Leu-Leu-norleucinal (ALLN), a protease calpain inhibitor, reduced RIPK1/RIPK3 expression and pMLKL via the signal transducer and activator of transcription 3 (STAT3) pathway. Blocking this calpain–STAT3–RIPK axis reduced ER stress and mitochondrial calcium dysregulation. In human lung transplant samples, mRNA levels of RIPK1, MLKL, and STAT3 were significantly increased at 2 h of reperfusion or post-transplant. Additionally, the levels of pRIPK1, pMLKL, and pSTAT3 are higher in human lung tissue samples from patients who developed PGD than those who did not [40]. The administration of Nec-1 to both donor lungs (preserved at 4 °C for 18 h) and recipients in a rat lung transplant model significantly improved pulmonary gas exchange, reduced lung edema, and inhibited necrosis [42]. Nec-1 has also been shown to significantly alleviate IR-induced lung injury, cytokine release, and the necroptosis of epithelial cells after left lung hilum clamping in mice [43]. Similarly, necrosulfonamide, an MLKL inhibitor, attenuated IR injury in a rat left hilum clamping and reperfusion model [44].

Wang et al. used a mouse left-lung transplant model to determine the donor lung conditions that may promote the development of PGD. In a single-hit model, donor lungs from inbred C57BL/6 mice were CI-preserved at 0 °C for 1 h, 72 h, or 96 h before engraftment. Multi-hit models were established by inducing 24 h of hemorrhage shock and/or 3 h of brain death before 24 h of CI preservation. Extending CI to 96 h led to increased necroptosis activation in lung grafts 24 h post-transplantation. Animals in the multi-hit group showed increased lung injury, cellular infiltration, and the activation of both necroptotic and apoptotic pathways. Nec-1 treatment significantly decreased necroptosis activation in both single- and multi-hit models of IR injury [45].

The initial recruitment of neutrophils to the reperfused lung is a critical step in IR injury hallmark post-transplantation. Using a mouse left-lung transplant model, Li et al. measured lipid peroxidation products with lipidomics and found that oxidized phosphatidylcholine species were rapidly increased after reperfusion [46], which is a signal associated with necroptosis [47]. The administration of Nec-1 to transplant recipients or using donor lungs from *Ripk3*^−/−^ mice significantly improved graft function by reducing neutrophil extravasation and aggregation, leading to better subpleural vessel integrity. Graft levels of oxidized phosphatidylcholine species were not elevated in RIPK3-deficient lungs [46].

## 4. Mechanisms of Necroptosis in Organ Transplantation

Organ transplantation may lead to the activation of several different and unique necroptotic pathways. Understanding these signalling pathways will aid in the development of pre-clinical and clinical strategies to prevent graft injury, dysfunction, and rejection.

### 4.1. DR-Dependent and Independent Activation of Necroptosis in Organ Transplantation

During organ transplantation, necroptosis can be activated via both extrinsic and intrinsic pathways. In the extrinsic pathway, TNFα and other inflammatory cytokines have been used to induce necroptosis in murine renal tubular epithelial cells (TECs) [20] and murine MVECs [23] in cell culture. Additionally, proinflammatory cytokines can up-regulate TNFR1 expression in murine MVECs [23]. Kidneys transplanted after CI preservation also have increased serum TNFα and the expression of TNFR1 and TLR4 during reperfusion [21]. Ischemic renal allograft transplant-induced remote lung injury in rats was associated with increased expression of TNFR1, TLR2, and TLR4 in lung tissue, and acute immune rejection increased TNFα levels in serum and lung tissue [36]. TNFR1 and TNFR2 double deficiency reduced graft arterial disease in murine cardiac allografts [48].

TLR3 can be activated by viral or endogenous RNA released from injured cells. In syngeneic cardiac transplants following 9 h CI, IR injury resulted in RNA release with the subsequent activation of TLR3 in the grafts. TLR3 inhibitor, or the use of cardiac grafts from Tlr3^−/−^ mice, reduced IR injury [49]. Furthermore, in murine lung transplantation, TLR9-dependent neutrophil extracellular trap formation is stimulated by mitochondrial DNA released from prolonged CI grafts [50] and may mediate neutrophil recruitment into ischemic tissue via a TLR9/MyD88/CXCL5 pathway [51]. The neutrophil rolling and adhesion to the vessel wall during reperfusion is mediated through TLR4/TRIF-dependent signalling in lung graft endothelial cells [51,52]. Considering these examples, the roles of different TLRs in the activation of necroptosis in IR injury and graft rejection should be further studied in organ transplantation settings.

Interestingly, Kim et al. used a cell culture model and demonstrated that neutralization antibodies against TNF, Fas, and TRAIL-R, during both CI and warm reperfusion, had no effect on cell death. Using TNFR1 siRNA to suppress its expression also did not reduce IR-induced cell death. The expression of cell death receptor interaction proteins, such as XIAP, cIAP, and caspase 8, were also not affected by Nec-1 treatment [40]. Therefore, the cold preservation and warm reperfusion conditions may activate necroptosis independent of DRs via the intrinsic pathway. In this setting, the acute depletion of ATP during CI may dysregulate sodium channels, depolarize the plasma membrane, and increase intracellular Ca^2+^ levels, leading to calpain activation and mediating cell death during reperfusion [53]. In lung epithelial cells, activated calpain has been shown to increase RIPK3 expression via the STAT3 pathway [40]. A newly discovered highly selective RIPK1 inhibitor, 6E11, protected human aortic endothelial cells from cold hypoxia–warm reoxygenation-induced cell death [54]. In MVECs, cold hypoxia and warm reperfusion enhanced TNFα- and IFNγ-induced necroptosis [27].

During organ transplantation, the enrichment of inflammatory cytokines and DAMPs in donor grafts may interact with multiple DRs to activate necroptosis. On the other hand, the IR conditions imposed on the donor organs may activate intrinsic necroptotic pathways directly. Therefore, to effectively block necroptosis, therapeutics targeting both intrinsic and extrinsic mechanisms should be considered (Figure 1).

### 4.2. Crosstalk between Necrosome and mPTP Opening in Transplant-Related Necroptosis

Mitochondria are the primary organelles for ATP and ROS production in the cell, and they also play a critical role in the regulation of cell death. The protein, p53, is a central stress sensor responsible for responding to multiple insults. The translocation of p53 to mitochondria and the formation of a complex with CypD trigger the opening of the mPTP and cause subsequent necrosis [17]. During ischemia, the mPTP remains closed and opens only once the tissues are reperfused after the ischemic period [55]. The induction of mPTP opening can cause excessive increases in intracellular Ca^2+^, leading to mitochondrial swelling and the rupture of the outer mitochondrial membrane [56,57]. The relationship between mPTP opening-regulated necrosis and necroptosis has been the subject of extensive research [58]; however, it requires further investigation in organ transplantation settings.

In MVECs, TNFα and caspase 8 inhibition-induced necroptosis was inhibited by mPTP inhibitor (S-15176), short-hairpin RNA (shRNA) for CypD, or cyclosporine A (which can bind CypD with high affinity to inhibit mPTP opening). Moreover, CypD deficiency or cyclosporine A inhibited the phosphorylation of MLKL in MVECs under necroptosis-inducing conditions [25]. Therefore, the IR conditions during the preservation and transplantation procedure may promote the interaction between necroptosis and mPTP opening-regulated necrosis. This crosstalk should be considered when developing effective therapies. The clinical application of cyclosporine A may protect the grafts from rejection through both immunosuppressive and anti-necroptosis pathways. Drugs with such dual roles or combining drugs with different roles may enhance the protection of the allograft during IR and/or rejection (Figure 1).

### 4.3. Relationship between Necroptosis, Apoptosis, and Other Types of Cell Death in Organ Transplantation

Both apoptosis and necroptosis have been identified as key injury hallmarks of the IR process in organ transplantation. They can both be activated by DRs, of which caspase activity is crucial in switching between these two types of cell death. As mentioned previously, caspase activation may negatively regulate necroptosis in a murine warm hepatic IR model [30]. In renal TECs [20] or in MVECs [23], TNFα induced necroptosis, as caspase activity was inhibited. In mice, caspase 8 silencing with shRNA in donor kidneys decreased renal allograft survival with increased tissue necrosis [20]. However, the inhibition of caspase 8 with shRNA attenuated IR injury induced by renal artery clamping at 32 °C in mice [59]. The conflicting roles of caspase 8 in IR injury necessitate further research to fully understand its dual role in cell death.

In a murine model, tubular cell apoptosis and caspase 9 expression were induced by CI, while the expression of both caspase 8 and necroptotic proteins (RIPK1, RIPK3, pMLKL, and TLR4) was significantly increased during reperfusion [21,39]. Similarly, apoptosis was observed after 18 h of CI in human lung epithelial cells, which was switched to necrosis after reperfusion [39]. Therefore, apoptosis and necroptosis may be involved in IR injury at different stages.

The pre-treatment of donors and recipients with a single dose of simvastatin 2 h prior to allograft transplant reduced the expression of mRNA and proteins for both apoptosis and necroptosis-related molecules in cardiac allografts after CI (4 h) followed by 6 h of reperfusion [29]. In a rat lung transplant model, apoptosis was observed after reperfusion from grafts preserved at a cold temperature for 6 h or 12 h, but necrosis was observed in grafts preserved at 18 h or 24 h [60]. Blocking apoptosis with a pan-caspase inhibitor [61] or annexin V dimer were both effective in reducing apoptosis [62] and improved graft function in lung transplantation. Thus, this work suggests that blocking apoptosis may prevent subsequent progression towards necroptosis. The administration of Nec-1 reduced CI–warm reperfusion-induced necroptosis as well as apoptosis in a cell culture model [43].

Taken together, these observations suggest that during organ transplantation under certain circumstances, apoptosis and necroptosis could co-exist or progress from one type to the other. The detailed mechanisms of these processes need to be co-investigated. Recently, a new concept, “PANoptosis” has been introduced—as an inflammatory PCD pathway with key features of pyroptosis, apoptosis, and/or necroptosis. It has been proposed that this is regulated by a so-called PANoptosome complex [63]. This and other types of PCD should be further studied in relation to one another to gain future insight into the temporal progression of injury in organ transplantation, which will be invaluable for determining the most effective regimen and time course of therapies.

## 5. Potential Therapeutic Targets and Substances Targeting Necroptosis

The inhibition of necroptosis represents a therapeutic strategy for IR injury and allograft rejection. Several substances have been used as experimental tools and could be further developed as therapeutics (Table 1).

### 5.1. RIPK1 Inhibitors

Necrostatins are a series of specific and potent small-molecule inhibitors of necroptosis. Necrostatin-1 (Nec-1) has been used in various solid organ transplantation models to block RIPK1 activity [20,23,24,40,42,43,45]. The new Nec-1 analog, necrostatin-1 stable (Nec-1s), showed improved pharmacokinetic properties in vivo [64], without any paradoxical toxicity at lower concentrations observed with Nec-1 [65]. Nec-1s was very effective in reducing necroptosis and the release of HMGB1 in cardiac graft injury and rejection [24,27].

GSK2982772 is an oral, selective, and specific inhibitor of RIPK1 [66]. Its safety and tolerability have been tested in healthy human volunteers (Phase 1) [67] and in a phase 2a clinical study for the treatment of inflammatory diseases [66,68]. 6E11, a highly selective inhibitor of RIPK1, shows more potent protective effects against cold hypoxia/warm reoxygenation-induced necroptosis in human aortic endothelial cells than Nec-1 or Nec-1s [54]. Zharp1-211, another novel RIPK1 inhibitor, selectively inhibits RIPK1 kinase activity and reduces graft-versus-host disease in mice [69].

### 5.2. RIPK3 Inhibitors

Three selective small-molecule compounds (GSK′840, GSK′843 and GSK′872) are shown to inhibit RIPK3-dependent necroptosis [70]. RIPK3 inhibitors have been shown to prevent cell death from a broader range of stimuli than RIPK1 inhibitors [71].

### 5.3. MLKL Inhibitor

Necrosulfonamide targets the pseudokinase domain of MLKL and has been shown to inhibit necroptosis in a rat pulmonary IR injury model [44].

### 5.4. CypD Inhibitor

Cyclosporin A, a calcineurin inhibitor, is a classic immunosuppressive drug for rheumatoid arthritis, psoriasis, Crohn’s disease, nephrotic syndrome, and organ transplants to prevent rejection [72]. It reduced necroptosis in MVECs through the inhibition of CypD-mediated mPTP opening [27].

### 5.5. Indirect Inhibitors for Necroptosis

Delta V1-1 (δV1-1), a specific peptide PKCδ inhibitor, ameliorated IR lung injury [39] and myocardial infarction [73]. N-acetyl-Leu-Leu-norleucinal (ALLN) is a strong, competitive inhibitor of the Ca^2+^-dependent neutral cysteine proteases calpain I and calpain II. It reduced IR-induced necroptosis in a cell culture model [40], IR-induced myocardial and cerebral injury [74,75], and IR injury in rat lung transplants [76].

However, navigating cell death treatment encounters certain challenges. Robust clinical evidence that firmly establishes the role of necroptosis in human organ transplantation is still lacking. Moreover, the utilization of drugs predominantly remains in experimental settings, as they are constrained by limited opportunities for clinical trials to comprehensively assess both safety and efficacy.

**Table 1 cells-12-02296-t001:** Substances that have inhibitory effects on necroptosis.

**Target**	**Substances**	**Research Setting**	**Models**	**References**
**RIPK1**	Necrostatin-1, Necrostatin-1 stable	Renal/heart/lung transplant	Cell culture and animal models	[20,23,24,40,42,43,45]
**RIPK1**	GSK2982772	Inflammatory diseases	Phase 2a clinical trial	[66,68]
**RIPK1**	6E11	Hypoxia/reoxygenation	Cell culture	[54]
**RIPK1**	Zharp1-211	Graft versus host disease	Animal model	[69]
**RIPK3**	GSK′840, GSK′843, GSK′872	Colon carcinoma	Cell culture	[70,71]
**MLKL**	Necrosulfonamide	Lung ischemia/reperfusion	Animal model	[44]
**Cyclophilin D**	Cyclosporin A	Heart transplant	Animal model	[27]
**Protein Kinase Cδ**	δV1-1	Lung transplant	Cell culture and animal models	[39,41]
**Calpain**	N-acetyl-Leu-Leu-norleucinal	Ischemia/reperfusion	Cell culture and animal model	[40,73,74,75]

## 6. Conclusions and Future Directions

### 6.1. Clinical Relevance of Necroptosis in Organ Transplantation

The causes of necroptosis in organ transplantation are complex (Figure 2). The incidence and severity of necroptosis are influenced by the donor conditions (e.g., age, fatty liver, hemorrhagic shock, brain death, circulatory death, etc.) and the systemic inflammatory responses, such as the crosstalk between the renal transplant and the lung [36,38]. The inflammatory responses in DBDs affect the quality of donor lungs [77]. A proteomics study revealed that biomarkers in the donor plasma of DBDs predict heart transplant outcomes [78]. Transcriptomics studies showed that cell death pathways are more predominately activated in DCD donor lungs than in DBD lungs [79]. Whether these pathways are related to necroptosis needs to be determined.

The organ preservation conditions also influence the degree of necroptosis. CI time differentially affects donor organs depending on their condition. Transcriptomic studies have demonstrated that inflammatory responses and cell death are the two predominant mechanistic pathways activated in human lung grafts during the IR process [77]. Similar gene expression changes in other solid organ grafts may also be studied with a focus on necroptosis.

Necroptosis is an important underlying mechanism for acute and chronic rejection after kidney [20] and heart [23,25,27,28] transplantation. Allografts from RIPK3-, CypD-, and TLR3-deficient donors exhibited prolonged graft survival. Similar studies should be conducted on other solid organ transplants.

Almost all of these studies are based on cell culture and animal models. There are only two human organ studies, one in the lung [40] and another in the liver [35]. More clinical research should be conducted to determine the significance of necroptosis in human organ transplants.

### 6.2. Molecular Mechanisms of Necroptosis in Organ Transplantation

Several unique features of necroptosis in organ transplantation are discussed in this review article. The activation of DRs by inflammatory responses and the activation of intrinsic pathways by IR conditions may have synergic effects in necroptosis. Crosstalk between the classical necroptotic pathways and mPTP opening-related pathways may accelerate the necrosis in grafts. Therapeutic strategies should be developed accordingly. Different types of cell death are associated with IR-induced graft damage and dysfunction in the kidney, liver, lung, and heart [80,81,82]. The relationship among different types of cell death needs to be studied further. Many cellular and molecular mechanisms of necroptosis have not been studied in organ transplantation, especially in clinical settings, and await exploration.

### 6.3. Therapeutics and Biomarkers of Necroptosis in Organ Transplantation

Several potential drugs targeting necroptosis were described in this paper. While most of these drugs have been used experimentally to elucidate the importance of necroptosis and potential mechanisms in the organ transplant setting, their clinical applications need to be developed systematically with careful pharmacological, pharmacokinetic, and pharmacodynamic and toxicity studies in pre-clinical animal models and then in clinical trials. Necroptosis-related biomarkers for clinical diagnosis and therapeutic guidance should, of course, be explored as well.

In conclusion, cell death plays a prominent role in the injury related to the transplantation of all organs. More extensive research into the mechanism of cell death and specifically necroptosis in organ transplantation is warranted. The examination of these pathways in clinical samples with clinical outcome data will reveal more opportunities for diagnosis and treatment in solid organ transplantation.

## Figures and Tables

**Figure 1 cells-12-02296-f001:**
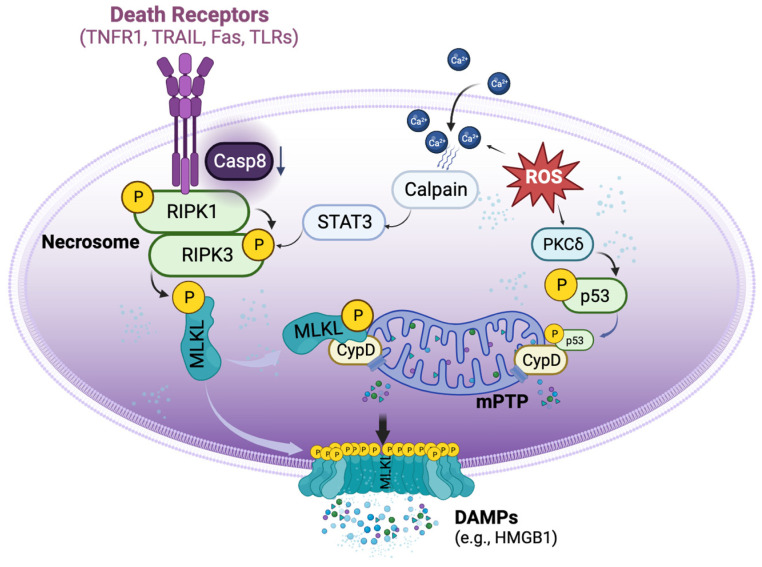
**Basic signalling pathways of necroptosis.** Necroptosis can be triggered by the activation of death receptors, which induce the phosphorylation of receptor-interacting protein kinase 1 (RIPK1) and RIPK3 to form the necrosome when the activity of caspase 8 is reduced. The necrosome then phosphorylates mixed lineage kinase domain-like protein (MLKL), causing it to translocate to the cell membrane and form pore-like structures that disrupt the integrity of the plasma membrane. These lead to the release of damage-associated molecular patterns (DAMPs), such as high mobility group box 1 (HMGB1). Aside from the activation of death receptors, necroptosis can also be triggered by reactive oxygen species (ROS). ROS can induce the translocation of p53 into the mitochondria, where it forms a complex with cyclophilin D (CypD), leading to the opening of the mitochondrial permeability transition pore (mPTP) and ultimately inducing regulated necrosis. Additionally, ROS can increase the accumulation of cytosolic Ca^2+^ and activate proteases such as calpain, which in turn activate necrosome formation.

**Figure 2 cells-12-02296-f002:**
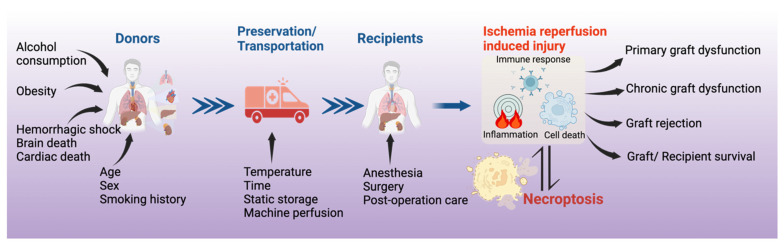
**The causes and significance of necroptosis in organ transplantation.** Necroptosis in the donor organ can be triggered by donor conditions and can further be affected by the conditions of organ preservation. Necroptosis contributes to ischemia–reperfusion injury and to primary and chronic graft dysfunction and rejection. Consequently, targeting necroptosis holds potential as a therapeutic strategy during different phases of organ transplantation.

## Data Availability

Not applicable.

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
