# Peer review of "Necroptosis in Organ Transplantation: Mechanisms and Potential Therapeutic Targets"

_cells, 2023, doi:10.3390/cells12182296_

Round 1

Reviewer 1 Report

This is a great review on cell death and organ transplantation. This manuscript reviewed and discussed the mechanism and effects of necroptosis on organ ischemia injury and organ transplantation. More importantly, the authors reviewed potential therapeutic targeting cell death in clinical transplantation. 

Overall, this is a very good review with adequate information. The finding contributes to the understanding of transplant tolerance. However, there are few aspects to be improved for better understanding this study.

Although detailed evidence for necroptosis in transplantation has been provided. It will be great if the authors can few sentences to discuss about potential challenge or drawbacks on targeting cell death treatments. In addition, for general readers easily following this review, it would be ideal to illustrate the cell death mechanism in the introduction.

Next are few minor suggestions:

1.      The authors discussed cold ischemia (CI) as a risk factor for acute kidney injury after kidney transplantation. After this, the authors reviewed the effect of hypothermic machine perfusion (HMP). Some connecting discussion or comments between above two aspects will be helpful for better understanding cell death and CI.

2.      There are other publications studied necroptosis in kidney injury and heart transplantation. It is better to cite their studies as well.

3.      It is a good idea to insert HMGB1 as a DAMP in the illustration.

Author Response

Reviewer #1

This is a great review on cell death and organ transplantation. This manuscript reviewed and discussed the mechanism and effects of necroptosis on organ ischemia injury and organ transplantation. More importantly, the authors reviewed potential therapeutic targeting cell death in clinical transplantation. 

Response: Thank you for your positive comments on our manuscript.

Overall, this is a very good review with adequate information. The finding contributes to the understanding of transplant tolerance. However, there are few aspects to be improved for better understanding this study.

Although detailed evidence for necroptosis in transplantation has been provided. It will be great if the authors can few sentences to discuss about potential challenge or drawbacks on targeting cell death treatments. In addition, for general readers easily following this review, it would be ideal to illustrate the cell death mechanism in the introduction.

Response: Following this suggestion, we have added a paragraph about potential challenges on targeting cell death treatment on page 9, line 410-414 in red font. Briefly, 1) lack of clinical evidence of necroptosis in organ transplantation, 2) most of drugs are used experimentally. The clinical trials for safety and efficacy are not available yet.

The cell death mechanisms have been illustrated in the Introduction (page 2, second paragraph and Figure 1).

Next are few minor suggestions:

  1. The authors discussed cold ischemia (CI) as a risk factor for acute kidney injury after kidney transplantation. After this, the authors reviewed the effect of hypothermic machine perfusion (HMP). Some connecting discussion or comments between above two aspects will be helpful for better understanding cell death and CI.

Response: This is also a great suggestion. We have added some sentences to explain the roles of cold ischemia and hypothermic machine perfusion in organ preservation on page 3, lines 106-108 and lines 118-121 in red font.

  1. There are other publications studied necroptosis in kidney injury and heart transplantation. It is better to cite their studies as well.

Response: We have found publications related to necroptosis in kidney injury and in other solid organ injury. However, this is not the focus of the present study.

Following this comment, we have searched PubMed again, to see if there is some new publication on necroptosis in heart or other organ transplantation. However, we did not find newer papers since our submission of this manuscript.

  1. It is a good idea to insert HMGB1 as a DAMP in the illustration.

Response: We have indicated HMGB1 as a DAMP in page 2, line 72, and have inserted it in figure 1.

Reviewer 2 Report

Cells 2568422

The main idea of ​​analyzing the role of necroptosis in organ transplantation sounds quite interesting, but unfortunately the authors do not work well on the subject and the mechanisms (promised in the title) are not clear after reading the manuscript.

In Introduction: Line 35 there is an extra space

Line 48 a la 57, add an extra phrase explaining the cellular and molecular mechanism of IR in the process of cell death

It is not so clear why necroptosis was chosen over another type of cell death. Need to better explain the subject.

Materials and Methods

Please inform the time-lapse and idiom criteria for the search. Why the authors choose those organs, what was the standards or criteria’s? indicate specifically how many original articles found by "organ", what were the exclusion criteria.

In Kidney, the authors only discuss CI, but what about other factors?  In line 164-165… please explain better the “role of caspases”, not only what Rosentreter reports, but a more exhaustive discussion. (This is because caspases are a relevant topic as the same authors mention later)

Line 190: a reference is loss

Line 212  the IR not the  …ischemia-reperfusion…

Since inflammation and in particular TNFa are relevant in necroptosis, these ideas need to be further developed. The same situation when discussing necroptosis, the authors must discuss in the context of other types of cell death, otherwise this work becomes a simple list of data.

Line 277-278 in the phrase: “The roles of these TLRs in activation of necroptosis”… the authors should to discuss here the role of they after to follow with the phrase “in IR injury and graft rejection should be further determined”

up to line 298 most of the information could have been summarized in one or two tables. This is because there is no great discussion or abstraction of the subject by the authors. The writing seems like a simple collection of data, without critical discussion or integration of the information.

Line 307-308, “However, the relationship between mPTP opening regulated necrosis and

necroptosis requires further investigation” It is hoped that the authors can provide sufficient information to explain the phenomenon. (there is enough information in pubmed)

In Table 1, a column is required where the mechanism (potential) of action is explained

Line 392 “The roles of necroptosis in organ transplantation are complex (Figure 2)” but figure 2 has no relationship with necroptosis. Please correct.

Please improve the wording and do not abuse the separate points. In all the manuscript, the writing does not follow a clear common thread.

Please improve the wording/grammatical rules and do not abuse the separate points. In all the manuscript, the writing does not follow a clear common thread.

Author Response

Reviewer #2

The main idea of ​​analyzing the role of necroptosis in organ transplantation sounds quite interesting, but unfortunately the authors do not work well on the subject and the mechanisms (promised in the title) are not clear after reading the manuscript.

Response: The cellular and molecular mechanisms of necroptosis have been the focus of much ongoing research. However, in organ transplantation research, many of these mechanisms have not been determined, especially in clinical setting. Based on our literature review and our own knowledge, we have discussed “Mechanisms of necroptosis in organ transplantation” (page 6- page 8). Following this critique, we added a sentence to indicate “Many cellular and molecular mechanisms of necroptosis have not been studied in organ transplantation, especially in clinical setting, and awaits further investigation". (Page 10, lines 448-450).

In Introduction: Line 35 there is an extra space

Response: The extra space has been deleted.

Line 48 a la 57, add an extra phrase explaining the cellular and molecular mechanism of IR in the process of cell death

Response: Thank you for your suggestion. We have added a sentence to explain IR-induced cell death. (New lines 55-57 in red font).

It is not so clear why necroptosis was chosen over another type of cell death. Need to better explain the subject.

Response: We have now explained “Among these PCD, necroptosis is particularly interesting, as it has been investigated in all major solid organ transplantation with clinical relevance”. page 2, line 59-61 in red font.

Materials and Methods

Please inform the time-lapse and idiom criteria for the search. Why the authors choose those organs, what was the standards or criteria’s? indicate specifically how many original articles found by "organ", what were the exclusion criteria.

Response: We have kept searching related paper until before submission of the paper (June 2023). We have now clarified this (page 2, line 89). Heart, liver, kidney, and lung are the major solid organs for transplantation. To clarify this, we have added “major solid organ” throughout the manuscript. We found 3 studies on necroptosis in kidney transplantation, 7 in heart transplantation, 1 in liver transplantation, and 6 studies on lung transplantation. In addition, 3 papers discussed kidney transplant-induced lung injury. All of these studies have been included.

In Kidney, the authors only discuss CI, but what about other factors?  

Response: In the kidney transplant, we discussed CI and HMP. There is a shortage of research on necroptosis in this area. Our goal with this review is to encourage more studies on necroptosis in various transplantation models.

In line 164-165… please explain better the “role of caspases”, not only what Rosentreter reports, but a more exhaustive discussion. (This is because caspases are a relevant topic as the same authors mention later)

Response: Thank you for this suggestion. We have revised the citation from Rosentreter’s paper (page 4, lines 171-174 in red font). More discussion on the relationship between necroptosis and apoptosis can be found from page 7 to page 8.

Line 190: a reference is loss

Response: A reference is added (#33). Line 186

Line 212  the IR not the  …ischemia-reperfusion…

Response: Change has been made as suggested. Line 221

Since inflammation and in particular TNFa are relevant in necroptosis, these ideas need to be further developed. The same situation when discussing necroptosis, the authors must discuss in the context of other types of cell death, otherwise this work becomes a simple list of data.

Response: The reviewer is correct. The inflammation, in particular TNFa are relevant in necroptosis. We further discussed this from page 6 to page 7, under “Death receptor-dependent and independent activation of necroptosis in organ transplantation”. In terms of relationship between necroptosis and other types of cell death, we have discussion from page 7 to page 8, under the subtitle of “Relationship between necroptosis, apoptosis and other types of cell death in organ transplantation”. Following your question, we have changed the subtitle a little bit to reflect your comments (page 7, lines 337-338).

Line 277-278 in the phrase: “The roles of these TLRs in activation of necroptosis”… the authors should to discuss here the role of they after to follow with the phrase “in IR injury and graft rejection should be further determined”

Response: Thank you for your suggestion. We have rephrased the sentence as “The roles of different TLRs in activation of necroptosis in IR injury and graft rejection should be further determined in organ transplantation settings. (Page 7, line 291-293).

up to line 298 most of the information could have been summarized in one or two tables. This is because there is no great discussion or abstraction of the subject by the authors. The writing seems like a simple collection of data, without critical discussion or integration of the information.

Response: Thank you for this comment. Necroptosis has been observed in different organs along various stages of the transplantation procedure (Ex. warm ischemia, cold ischemia, machine perfusion, post-transplant reperfusion etc.). As such, original research is usually confined to a specific organ system and different points in the overall injury cascade. We decided to report these results rather than create a table because of the complexity between models, organ system, time of measurements etc. We believe that using this reporting style will more thoroughly inform readers by allowing them to direct their attention to areas of interest.

Line 307-308, “However, the relationship between mPTP opening regulated necrosis and necroptosis requires further investigation” It is hoped that the authors can provide sufficient information to explain the phenomenon. (there is enough information in pubmed)

Response: The reviewer is correct. There are many publications on the interaction between mPTP opening regulated necrosis and necroptosis. We have pointed this out to the reader (lines 321-324) and provided a review article as a reference should they want more information.

In Table 1, a column is required where the mechanism (potential) of action is explained

Response: The mechanism can be inferred via the “Target” column. We avoided adding a “mechanism (potential) of action” column to reduce redundancy and for many of the compounds listed the complete mechanism is still an area of research.

Line 392 “The roles of necroptosis in organ transplantation are complex (Figure 2)” but figure 2 has no relationship with necroptosis. Please correct.

Response: This is a valid critique. We have changed the title of Figure 2 and added a sentence in the figure legend to reflect its relationship to necroptosis.

Please improve the wording and do not abuse the separate points. In all the manuscript, the writing does not follow a clear common thread.

Response: We have had all our co-authors revise the manuscript to improve delivery. We hope the reviewer can appreciate the changes made.

Reviewer 3 Report

Authors summarized the scientific literatures on the prevalence of necroptosis in solid organ transplantation, focused on the regulatory mechanisms of necroptosis discovered from organ transplantation studies. Furthermore introduced potential therapeutic targets for necroptosis, and proposed future directions on PCD related research in organ transplantation.

Minors:

- please state cleraly if this review is systematic (vs narrative). If systematica several methodological informations should be added

- timing is crucial in donor's assessment, to make the process safe and to match donors and recipients. There is a bottleneck in this process you have to discuss and related to the role of pathological investigation of suspicious lesions or organs to transplant. In this case potential dramatic consequences lilke transmission can happen, please discuss and quote (PMID: 28665524, PMID: 24034231,PMID: 11602897).

- furthermore considering timing we need to consider tools that can make easy, fast and safe the process. In this context technological innovations are a tool. Please discuss and quote (PMID: 35941875, PMID: 11774166)

English is correct and readable

Author Response

Reviewer #3

Authors summarized the scientific literatures on the prevalence of necroptosis in solid organ transplantation, focused on the regulatory mechanisms of necroptosis discovered from organ transplantation studies. Furthermore introduced potential therapeutic targets for necroptosis, and proposed future directions on PCD related research in organ transplantation.

Response: Thank you for your positive comments on our manuscript.

Minors:

- please state cleraly if this review is systematic (vs narrative). If systematica several methodological informations should be added

Response: Thank you for your suggestion. We have clarified that this is a narrative review (page 2, line 85).

- timing is crucial in donor's assessment, to make the process safe and to match donors and recipients. There is a bottleneck in this process you have to discuss and related to the role of pathological investigation of suspicious lesions or organs to transplant. In this case potential dramatic consequences like transmission can happen, please discuss and quote (PMID: 28665524, PMID: 24034231,PMID: 11602897).

Response: Thank you for your suggestion. We have read the publications you have suggested related to cancer transmission from donor to recipient. We have added this important consideration (page 1, lines 44-45). However, we opted to not include these three publications to our references because we just briefly mention carcinogenesis in organ transplantation as it is not the focus of this review. From lines 43-46 we detail several challenges along with carcinogenesis that should be considered, but can be explored by the reader further in different reviews.

- furthermore considering timing we need to consider tools that can make easy, fast and safe the process. In this context technological innovations are a tool. Please discuss and quote (PMID: 35941875, PMID: 11774166)

Response: We thank the reviewer for the recommendation; however, technological innovations, as those referred to in the publications listed, are beyond the scope of our review. We agree that these are important aspects to transplantation medicine and would be better suited to the topic in another review.

Round 2

Reviewer 2 Report

The authors made most of the requested changes. The writing was understandable and more logical, even so it is necessary that up to line 335 information should summarize in one (or two) tables. This is because there is no great discussion or abstraction of the subject by the authors. The writing seems like a simple collection of data, without critical discussion or integration of the information.

Author Response

The authors made most of the requested changes. The writing was understandable and more logical, even so it is necessary that up to line 335 information should summarize in one (or two) tables. This is because there is no great discussion or abstraction of the subject by the authors. The writing seems like a simple collection of data, without critical discussion or integration of the information.

Response: Thank you for accepting most of our revision. In the previous round, you also suggested us to summarize our text to one or two Table(s) for the same reason. However, even though necroptosis has been observed in different organs along various stages of the transplantation procedure, the numbers of publications in each organ is limited. The roles of necroptosis have been started by different groups based on their own interests. It is not a systemic exploration for this important subject at this stage. The studies reviewed here should be considered as a showcase. To briefly introduce their observations in the main text will show good examples to promote further research on necroptosis in organ transplantation. We, respectfully, did not make further changes for our manuscript.